# Fractionation and Extraction Optimization of Potentially Valuable Compounds and Their Profiling in Six Varieties of Two *Nicotiana* Species

**DOI:** 10.3390/molecules27228105

**Published:** 2022-11-21

**Authors:** Csaba Laszlo, Kacper Kaminski, Haifeng Guan, Maria Fatarova, Jianbing Wei, Alexandre Bergounioux, Walter K. Schlage, Sandra Schorderet-Weber, Philippe A. Guy, Nikolai V. Ivanov, Kai Lamottke, Julia Hoeng

**Affiliations:** 1PMI R&D, Philip Morris Products S.A., Quai Jeanrenaud 5, CH-2000 Neuchâtel, Switzerland; 2Natural Product & Drug Discovery Department, Bicoll Biotechnology (Shanghai) Co., Ltd., Bibo Road 518, Zhangjiang Science City, Pudong, Shanghai 201203, China; 3Biology Consultant, Max-Baermann-Str. 21, 51429 Bergisch Gladbach, Germany; 4Bicoll GmbH, Am Klopferspitz 19, 82152 Planegg, Germany

**Keywords:** extraction, purification of plant compounds, microfractionation, *Nicotiana tabacum*, *Nicotiana rustica*, valuable compound profiling

## Abstract

There is an increasingly urgent call to shift industrial processes from fossil fuel feedstock to sustainable bio-based resources. This change becomes of high importance considering new budget requirements for a carbon-neutral economy. Such a transformation can be driven by traditionally used plants that are able to produce large amounts of valuable biologically relevant secondary metabolites. Tobacco plants can play a leading role in providing value-added products in remote areas of the world. In this study, we propose a non-exhaustive list of compounds with potential economic interest that can be sourced from the tobacco plant. In order to optimize extraction methodologies, we first analyzed their physico-chemical properties using rapid solubility tests and high-resolution microfractionation techniques. Next, to identify an optimal extraction for a selected list of compounds, we compared 13 different extraction method–solvent combinations. We proceeded with profiling some of these compounds in a total of six varieties from *Nicotiana tabacum* and *Nicotiana rustica* species, identifying the optimal variety for each. The estimated expected yields for each of these compounds demonstrate that tobacco plants can be a superior source of valuable compounds with diverse applications beyond nicotine. Among the most interesting results, we found high variability of anatabine content between species and varieties, ranging from 287 to 1699 µg/g. In addition, we found that CGA (1305 µg/g) and rutin (7910 µg/g) content are orders of magnitude lower in the Burley variety as compared to all others.

## 1. Introduction

Tobacco was used by Native Americans for ceremonial and medicinal purposes [1]. “Traditional” tobacco is known today as the *Nicotiana rustica* species. It was originally consumed by chewing, in a manner similar to coca leaves. This custom was gradually adopted by European settlers and spread across multiple continents. Another indigenous way of consuming tobacco—sniffing pulverized leaves—was also gradually adopted and became popular. The most popular method of tobacco consumption, however, became smoking. Although it was nicotine that drove the popularity of tobacco, there is a complex biosynthetic mechanism that leads to the production of many different compounds in tobacco plants. Common tobacco (*Nicotiana tabacum* L.) is primarily associated today with consumer goods production; however, similar to many other plants, it also efficiently produces a variety of valuable secondary metabolites. Therefore, tobacco species and the accumulated knowledge regarding their agriculture make them a potential keystone of the bio-economy transformation process as a non-food source, a CO_2_-negative resource applicable at an industrial scale [2]. 

*N. tabacum* L. is an allotetraploid plant with a 4.5-gigabase (Gb) genome, one of the largest and most gene-rich among all common crops, roughly five times larger than potato and tomato [3]. With 90,000 genes, tobacco may possess superior potential for producing biologically complex compounds that are readily available on a large scale. It is estimated that all tobacco tissues contain ~5700 different chemical constituents including carbohydrates, nitrogenous compounds, alkaloids, pigments, isoprenoids, carboxylic acids, phenolics, sterols, and inorganic compounds [4]. Figure 1 depicts these compounds classified according to their biosynthetic pathway. 

### 1.1. Alkaloids

There are four major secondary alkaloids that accumulate in the *Nicotiana* genus: nicotine, nornicotine, anabasine, and anatabine. These alkaloids constitute the vast majority of the total alkaloid pool [9,10]. For many of these species, nicotine is the major alkaloid accumulated (e.g., in common tobacco—*N. tabacum* L.), which typically accumulates up to 4% of the total dry weight. Nicotine accounts for >90% of the pool in the *Nicotiana* genus, with nornicotine, anatabine, and anabasine making up the majority of the rest of the alkaloid pool. While nicotine and nornicotine biosynthesis is well described in the literature, anatabine- or anabasine-specific enzymatic steps have been barely characterized [11].

The two heterocyclic rings that form nicotine originate from separate amino acids: the pyridine moiety is derived from aspartic acid, and the pyrrolidine ring from ornithine or arginine. The biosynthesis of the pyridine ring starts with the oxidation of aspartate to α-iminosuccinate, which is then converted to quinolinic acid and is further transformed to nicotinic acid mononucleotide (NaMN) [12,13]. Nicotinic acid—the building block contributing the pyridine ring to the structure of nicotine—is then formed from NaMN. Pyrrolidine ring formation proceeds via the polyamine putrescine, which can be formed in two different pathways: either by direct conversion of the non-proteinogenic amino acid ornithine to putrescine [14,15] or from arginine through a three-step process [16]. Putrescine is then converted to *N*-methyl putrescine [17,18] and further to *N*-methyl aminobutanol [19,20]. *N*-methyl aminobutanol is assumed to cyclize spontaneously to form the *N*-methyl-Δ’-pyrrolinium ion, a direct substrate for the last step of nicotine biosynthesis. For the final steps in nicotine biosynthesis, nicotinic acid is converted to 3,6-dihydronicotinic acid [17,21,22]. It has been suggested that 3,6-dihydronicotinic acid is decarboxylated to 1,2-dihydropyridine and 2,5-dihydropyridine, the former of which reacts with the *N*-methyl-Δ’-pyrrolinium ion to form nicotine [23]. Finally, nicotine can be converted to nornicotine, the only alkaloid that is also formed in tobacco leaves in addition to the roots [24,25,26].

Anatabine biosynthesis differs significantly from that of nicotine, as both of its rings originate from the pyridine-nucleotide pathway, once nicotinic acid is converted to 3,6-dihydronicotinic acid. Although no anatabine-specific enzymes have been discovered, radiolabeling biomimetic studies suggest that it is further decarboxylated to 1,2-dihydropyridine and 2,5-dihydropyridine [27]. These two intermediates can either react together to form 3,6-dihydroanatabine, or the same effect can occur from dimerization of 2,5-dihydopyridine itself. In the previously proposed further step of biosynthesis, 3,6-dihydroanatabine is hydrogenated to form anatabine [28,29]. 

Anabasine is structurally similar to nicotine, where its pyridine ring is believed to be the same as that of nicotine [30]; it also consists of a piperidine ring derived from cadaverine. While for nicotine biosynthesis, it is putrescine that is produced from ornithine, for anabasine, it is cadaverine derived from lysine [11,31,32]. Cadaverine is subsequently deaminated by diamine oxidase to 5-aminopentanal, which is then believed to spontaneously cyclize to Δ1-piperdeine, a substrate that together with nicotinic acid forms anabasine. 

Myosmine was found in previous pyridine alkaloid content studies of *Nicotiana* species, although in minute quantities [33,34]. Sun et al. (2013) detected and quantified myosmine in leaves of *N. tabacum* varieties and *N. tomentosiformis* [35]. Myosmine itself is considered to arise from nornicotine degradation [36,37]. Finally, although cotinine is a major metabolite of nicotine breakdown in humans, its biosynthesis remains uncharacterized possibly due to its very low concentrations in various *Nicotiana* species [38]. Nicotinic acid and nicotinamide—known as vitamin B3 components—are not only of paramount importance to alkaloid biosynthesis but together with tryptophan are key to NAD+ biosynthesis, an indispensable molecule for cellular energy metabolism [39,40].

### 1.2. Rutin

Flavonoids and anthocyanins are responsible for the color of flowers, fruits, and sometimes leaves. They are also important as secondary metabolites that protect tobacco from damage from reactive oxygen species (ROS) [41]. Flavonoids are synthetized from coumaroyl-coenzyme A (CoA) to form naringenin chalcone, which is then converted to naringenin (C6-C3-C6 structure) [42] and subsequently to specific flavonoids and anthocyanins. The very first step in flavonol and anthocyanin synthesis is the conversion of naringenin to dihydrokaempferol, which is subsequently converted to dihydroquercetin [43]. These two compounds are then transformed to kaempferol and quercetin, [43]. In the last steps of anthocyanin synthesis, leucocyanidin and leucopelargonin are converted to cyanidin and pelargonin. More than 4000 known flavonoids have been identified in tobacco [44], with the most predominant being rutin (nearly 8 mg/g), astragalin, isoquercetin, and karacyanin [9]. 

Most flavonoids are assumed to protect tobacco from ROS, which increase dramatically during ultraviolet (UV) radiation. This is one of the most obvious production improvements that could be applied. It has been shown that phytochrome B is directly involved in the regulation of flavonoid production. Production of phenolic compounds can also be increased by chemical elicitors, pressure, electric field, heavy metals, pH change, or temperature increase. Microorganisms, fungi, and bacteria can also enhance phenolic compound accumulation [45]. 

Nearly all flavonoids have potential use and are studied for their antioxidant, anticancer or antimicrobial properties, as well as for cardiovascular disease management [44,46]. With wide and increasing applications in various chemical, food, and medical industries, rutin is one of the most important flavonoids that occurs in nature. Tobacco can accumulate up to 0.8% rutin in its leaves [7], most of which can be readily extracted. Rutin is one of the most effective antioxidants present in nature. It is synthetized in higher plants such as various citruses but also to a very high degree in tobacco. Patel and Patel [47] in an excellent review book state that it has “various pharmacological activities such as antibacterial, antiprotozoal, antitumor, anti-inflammatory, antiallergic, antiviral, cytoprotective, vasoactive, hypolipidaemic, antiplatelet, antispasmodic, and antihypertensive. Rutin is used in food in different forms such as colorant, antioxidant, preservative, stabilizer, and UV absorbent. It is also used as an active component in various herbal medicines, multivitamin preparations, the cosmetic and chemical industries, and in animal feed”. Uses of rutin to treat several key medical conditions such as Parkinson’s disease, Alzheimer’s disease, dementia, or ulcerative colitis have been patented and are being applied [48]. Oral, parenteral, topical, and inhalation are the preferred administration routes for rutin and its derivatives. Rutin is also used in combination with ferulic acid (FA) for neurodegenerative and gastrointestinal diseases, while rutin itself is used in combination with nicotinic acid for constipation and neurodegenerative diseases.

### 1.3. Chlorogenic Acids

Chlorogenic acid (CGA) is an active dietary polyphenol with a wide range of potential health benefits. CGA has anti-diabetic, anti-carcinogenic, anti-obesity, neuroprotective, and anti-hypertensive properties [49,50,51]. Additionally, CGA acts as a strong antioxidant, is reported for its antibacterial properties, can inhibit lipogenesis, and has anti-inflammatory activities. Believed to improve skin appearance, counteract skin aging, and used for the treatment of acne vulgaris, it is a compound of interest for the cosmetic industry [52,53].

Tobacco can accumulate CGA in leaves up to 2.4% of dry weight, making this plant one of the best sources [54]. CGA is the ester of caffeic acid and the (-) form of quinic acid. However, CGA refers to an entire family of polyphenols, members of which are formed through esterification of feruloyl-, dicaffeoyl-, and coumaroylquinic acids with caffeic acid. CGAs can be found in raw coffee, sunflower seeds, and blueberries, with lower contents in potatoes, tomatoes, apples, pears, and eggplants [49]. Chemical-based assays have shown that CGA has the capacity to scavenge radicals, superoxide anions, and peroxynitrite, as well as protect against lipid oxidation and hydrogen peroxide-induced DNA plasmid and chromosome breaks [52]. Cell-based assays confirmed the antioxidant capacity of CGA after various chemical stimuli. This antioxidant effect was also attributed to improvements in different in vivo disease models. For example, an animal model of Alzheimer’s disease showed decreased malondialdehyde levels in both the frontal cortex and hippocampus [55]. The anti-amnesic activity of CGA was attributed to reduced lipid peroxidation in addition to increased free radical scavenging activity. Pretreatment of guinea pig dorsal skin with a CGA-containing oil-in-water type-microemulsion gel prevented UV-B-induced erythema formation. It was suggested that CGA protects the skin against UV-induced oxidative damage [56]. Oral administration of CGA to mice before exposure to gamma radiation significantly reduced chromosomal damage [57]. To note, photo-oxidation protection in a mouse epidermal cell line and human HaCaT keratinocytes was related to the induction of nuclear factor erythroid 2-related factor 2 (Nrf2) transactivation and phase II enzyme activities [52]. Wound healing was accelerated in diabetic-induced rats after intraperitoneal injection of CGA, and the wound beds had decreased malondialdehyde and nitric oxide levels with higher reduced-glutathione [57]. In induced-obese mice, CGA supplementation with a high-fat diet significantly lowered body weight, visceral fat mass, plasma leptin and insulin levels, and concentrations of triglycerides (in plasma, adipose tissue, liver and heart) and cholesterol (in plasma, adipose tissue and heart) compared to the high-fat diet control group [58]. Plasma adiponectin was also elevated by CGA supplementation compared to the high-fat diet control group. CGA significantly inhibited fatty acid synthase, 3-hydroxy-3-methylglutaryl CoA reductase, and acyl-CoA: cholesterol acyltransferase activities, and increased fatty acid beta-oxidation activity and peroxisome proliferator-activated receptors alpha expression in the liver, suggesting improved lipid metabolism [58].

Many other studies have provided evidence for the anti-inflammatory activity of CGA. Intradermal injection and oral administration reduced neutrophil infiltration and inhibited the nuclear factor NF-κB pathway in an induced colitis mice model [59]. In another colitis model, CGA attenuated dextran sulfate sodium-induced body weight loss, diarrhea, fecal blood, and colon shortening; dramatically improved colitis histological scores; and downregulated pro-inflammatory cytokine levels [60]. In a rat carrageenan-induced paw edema model, suppression of pro-inflammatory cytokines was also observed [61].

### 1.4. Ferulic Acid

FA is a major phenolic acid in the tobacco plant that is widely used in the pharmaceutical, food, and cosmetics industries. It has low toxicity and improves many pathophysiological conditions via its anti-inflammatory and antioxidant-inducing properties. Nonclinical evidence demonstrated antimicrobial activity, anticancer, hepatoprotective, cardioprotective, neuroprotective, and antidiabetic effects [62,63]. A clinical study reported reductions of cardiovascular risk factors in hyperlipidemic patients treated with FA (1 g/day) [64]. In a study sponsored by Philip Morris International (PMI), FA improved memory functions in mice [65]. It is also applied in skin care formulations as a photoprotectant and anti-aging component [63,66].

FA is a hydroxycinnamic acid and is closely related to other phenolic acids including CGA, p-coumaric acid, and caffeic acid [67]. It is an important constituent of plant structural biomass. This monomeric precursor of lignin stabilizes cell walls, and it can act as an intermediate metabolite for soluble pant materials (e.g., a one-step chemical reaction can convert FA into vanillin) [68]. High levels of FA are found in both free and bound forms in vegetables, fruits, cereals, and coffee (e.g., 0.485 mg/g dry weight in rice, 1.213 mg/g dry weight in buckwheat [69], and the highest reported concentration of 33 mg/g fresh weight in refined corn bran) [67]. It has been estimated that consumption of these foods may result in approximately 150–250 mg/day of FA intake [67]. A wide range of FA content has been reported for the tobacco plant: 30 to 60 µg/kg fresh weight (leaves) and 10-times higher in roots [70]. Simple alkaline or solvent extraction methods are effective for FA isolation.

FA’s mode of action includes activity as a free radical scavenger, but it also inhibits enzymes that catalyze free radical generation, and it enhances scavenger enzyme activity. In vitro, FA can suppress the pro-inflammatory NF-κB and mitogen-activated protein kinase pathways, and it modulates the activities of and crosstalk between NF-κB and Nrf-2 signaling [71]. Moreover, it was shown to promote wound healing in gingival fibroblasts exposed to heavy metal-induced oxidative stress when co-administered with other compounds [72,73]. In vivo, FA alleviated lipopolysaccharide (LPS)-induced acute respiratory distress syndrome in mice through its anti-inflammatory and antioxidant activities [74]. In a rat model of pre-eclampsia, FA reduced the inflammatory response [75]. The cardiotoxic side effects of long-term anticancer doxorubicin treatment, caused by oxidative stress, were ameliorated by FA stimulation of the Nrf2 pathway, leading to enhanced synthesis of major Nrf2-dependent antioxidants [76]; similar effects were reported against liver damage following methotrexate chemotherapy [77].

### 1.5. Zeatin

Zeatin is an extractable plant hormone, belonging to the class of cytokinins, that can be isolated from a variety of plant materials including tobacco in the ng/g range [78]. Prices are in the range of 6000 EUR/g (https://www.sigmaaldrich.com/DE/en/product/sigma/z0164) as of 20 November 2022. Zeatin has become a strongly patent-covered anti-aging ingredient of various cosmetics/skin care products [79,80]. It is also used for dietary supplements and attracts interest in medical applications based on its potential efficacy in diabetes and cognitive diseases [81,82,83]. Regarding crop protection, endogenous zeatin has a broad functional spectrum in plant physiology including the activation and control of stress responses and plant immunity against microbial infections. The question of whether extracted zeatin added externally to crops such as tobacco plants can be efficiently used to enhance their resistance against microbial pathogens is still under investigation.

Grosskinsky and coworkers investigated plant hormone interactions in tobacco (*N. tabacum*) plants induced by the pathogenic bacterium *Pseudomonas syringae*. These researchers initially used gene transfection to increase cytokinin synthesis in response to pathogen infection, showing that cytokinins mediate enhanced resistance against the tobacco pathogen *P. syringae* pv. tabaci. The cytokinin-mediated resistance strongly correlated with an increased level of bactericidal activity and upregulated synthesis of the two major antimicrobial phytoalexins in tobacco: scopoletin and capsidiol [84]. 

Zeatin exists in *cis*- and *trans*-configurations (with differing activities, see below) and can act in the form of zeatin riboside. There is evidence that externally applied *trans*-zeatin in concentrations around 10 micromolar can efficiently reduce *P. syringae* replication for up to 1 week post treatment, while *cis*-zeatin can increase tobacco tolerance to the infection and thereby reduce the symptoms [85]. It was also shown that endogenous defense mechanisms against *P. syringae* can be activated by a pretreatment with disarmed *Agrobacterium tumefaciens*, which synthesizes its own *trans*-zeatin, thereby triggering tobacco’s cytokinin mechanism. In addition to antimicrobial plant stimulation, zeatin can increase resistance of tobacco against damaging insects including the tobacco hornworm [86].

### 1.6. Linoleic Acid

Linoleic acid (LA) is an essential fatty acid and must be obtained from external food sources. LA is being studied in relation to multiple human diseases such as cancer, obesity, and atherosclerosis [87]. Reduced low-density lipoprotein cholesterol levels and a lower risk of hypertension have been associated with higher LA dietary intake [88]. It has been proposed that the conjugated form of LA (CLA) is responsible for some of these health benefits [89].

### 1.7. Neopytadiene

Terpenes are a class of natural products predominantly produced by plants. They can be classified depending on the number of carbons as mono- (C_10_, two isoprene units), sesqui- (C_15_, three isoprene units), di-terpenes (C_20_, four isoprene units) and so on. They are primary building blocks of numerous plant molecules such as steroids (derived from squalene, a tri-terpene). Among other functions, they play roles in plant defense and disease resistance and can be used as agricultural pesticides [90]. A variety of industrial products such as varnishes, natural rubber, a polyisoprene, flavors, perfumes, cosmetics are derived from terpenes [91].

Neopytadiene is a diterpene that is accumulated in high content in tobacco (up to 0.25% of dry weight). Total content increases during yellowing and the curing process [92,93]. A previous study detected antimicrobial activity in plant essential oil containing neophytadiene [94]. Neophytadiene is also reported as a treatment for headache, rheumatism, and some skin diseases [95]. In a recent 2020 study, neophytadiene isolated from marine algae showed anti-inflammatory effects during LPS induction both in macrophages and rats [96]. 

### 1.8. Experimental Approach

In the current study, we employed a microfractionation technology called Bifrac N™ that provides high-resolution separation of analytes. This separation technology is applicable even to complex plant matrices, is readily available, and has proven efficiency with various plant samples, especially for comparing different plant species and organs. Such microfractionation techniques open the door to bioactivity-guided small molecule isolation as a major research tool for identifying novel molecules with potential pharmacological applications [97]. This approach combines reliable separation methods for natural products with innovative new techniques, by sorting the complex matrix of compounds existing in the starting materials (crude plant extracts) based on their physicochemical properties (logP, solubility). The obvious advantages are the gentle and efficient separation conditions. Compared with traditional separation technologies (e.g., via silica gel or octadecylsilane), it greatly minimizes the loss of sample complexity, especially for minor compounds with potential biological activities. General separation techniques often exhibit irreversible adsorption or cannot access small molecules of interest due to their inherent solubility issues [98]. Using this approach for *N. tabacum* and *N*. *rustica* varieties enables much faster and safer access to various small, bioactive molecules from natural resources. Moreover, it facilitates the quantification of biologically relevant secondary metabolites. Each well is normalized to a defined amount (e.g., 0.2 mg), making them readily available for high-throughput screening demands. This microfractionation technology generates up to 192 fractions per plant sample, with reduced complexity (e.g., containing 3–6 major compounds per microfraction). 

This approach largely avoids false positives, false negatives, and loss of activity in biological screening during the course of further purification to the single active pure compound. Considering a broad range of applications including biomolecular assays, whole cell assays, and whole organisms, this technology offers a valid starting point for pharmaceutical, nutraceutical, cosmetic and agricultural fields [99].

Microfractionation followed by functional assays has been successfully applied to high-throughput screening and identification of the natural product neoruscogenin as a bioavailable, potent, and high-affinity agonist of the nuclear receptor RORα (NR1F1) RORα [100], as well as the deconvolution of *Cistus incanus* extracts showing potent and broad in vitro antiviral activity against human immunodeficiency virus and filoviruses, by targeting viral envelope proteins [97].

In this study, the microfractionation technology Bifrac N™ was employed for pre-treatment and enrichment of tobacco plant extracts for easier detection before quantification of potentially valuable compounds buried in the complex plant matrix.

## 2. Results

### 2.1. Rapid Solubility Test of Potentially Valuable Compounds

Determination of the small molecule content in a complex plant matrix depends on the understanding of minimum solubility in the chosen extraction solvent. Usually, it is not reported in the literature if the disclosed method of choice is the best approach for the compound(s) of interest. This is especially important if the compound of interest has a variable solubility depending on its purity. Often, tests on mixture are hardly reproducible on pure compounds if this aspect is not routinely considered in the workflow. We performed a general testing of best solvent systems to compare a wide range of small molecules of interest to understand the limitations of general approaches within a complex plant matrix such as *Nicotiana.* First, the solubilities of 29 pure reference compounds were tested in 7 solvents of different polarities ranging from nonpolar to polar. The concentrations of the compounds varied between 0.19 and 558 mg/mL. As expected, the results in Table 1 indicate that no single polar or nonpolar solvent could simultaneously cover the solubilities of all the selected reference compounds of interest in an optimal fashion (good solubility defined as >100 mg/mL). Hexane (C_6_H_14_) showed good solubility for 13 out of 29 reference compounds, dichloromethane (CH_2_Cl_2_) for 18/29, dichloromethane–methanol (CH_2_Cl_2_/MeOH; 4:1, *v*/*v*) for 20/29, ethanol (EtOH) for 13/29; methanol (MeOH) for 13/29, methanol/water (MeOH/H_2_O; 70:30, *v*/*v*) for 6/29, and water (H_2_O) showed good solubility for 5/29 reference compounds. From these results, the best solvents were ranked as (1) CH_2_Cl_2_/MeOH (4:1, *v*/*v*), (2) CH_2_Cl_2_, (3) EtOH, (4) MeOH, (5) C_6_H_14_, (6) MeOH/H_2_O (70:30, *v*/*v*), and (7) H_2_O. The mixed solvent of CH_2_Cl_2_/MeOH (4:1, *v*/*v*) covered the good solubility of most (69%) of the chosen, pure (purity of standards +90%) reference compounds, except for anabasine, nicotinamide, nicotinic acid, zeatin, rutin, CGA, ferulic acid, zeaxanthin and lutein that showed <100 mg/mL solubility.

Although rapid solubility tests were realized from reference compounds, the results highlight the importance of the plant matrix in terms of complex sample and pH associated to the plant starting material. As an example, rutin was not very soluble in our fast solubility assays while high concentration levels were obtained using solvent extraction (Table 2 and Table 3). A majority of the compounds were better extracted from polar extracts, except for cotinine, norcotinine, LA, α-tocopherol, squalene, and neophytadiene (higher concentration levels found in apolar extract fractions). Finally, nicotinamide and nicotinic acid results were not so conclusive between the polar or apolar extraction fractions. 

It has been envisioned that by changing the ratio of CH_2_Cl_2_ and MeOH, the solubility of reference compounds, even the poorly soluble ones, would be improved. Thus, the solvents CH_2_Cl_2_, CH_2_Cl_2_/MeOH (4:1, *v*/*v*), CH_2_Cl_2_/MeOH (1:1, *v*/*v*), CH_2_Cl_2_/MeOH (1:4, *v*/*v*), MeOH, and 90% MeOH/H_2_O were chosen as the favorable/proposed extraction solvents for further extraction optimization.

### 2.2. Coverage by High-Resolution Microfractionation

A set of 96 microfractions originating from a nonpolar crude extract (obtained with dichloromethane (DCM) and ethyl acetate (EtOAc) extraction) and the corresponding set of 96 microfractions of a more polar crude extract (obtained with 95% EtOH extraction) were used for screening. Due to the microfractionation procedure and the sequential screening of each fraction in the series, the signals measured in the neighboring wells form a continuous, bell-shaped distribution of the activity (or compound concentration), resulting in active fraction clusters. This is different from a random distribution pattern, usually observed in primary screening from synthetic small molecule compound libraries and is an inherent checkpoint for the quality of an assay system. These fractions are the source of qualified and biologically relevant small molecule drug candidates with defined properties. 

To test the applicability of microfractionation, the total amounts of 24 compounds of interest were quantified by liquid chromatography (LC) in three different *N. tabacum* varieties: Virginia (V), Burley (B), and Oriental (O) (Table 2). The cumulated contents are representative of the specific fraction cluster where the compound was detected. The concentrations ranged from 0.1 to 2277 µg/g dry material except for one compound (vitamin D_3_, which was not detected). The abundant compounds whose contents were above the level of 200 µg/g dry material are as follows (µg/g dry material): rutin (2277 in V, 2129 in O), CGA (1574 in V, 1109 in O), nicotine (1245 in V, 683 in B, 379 in O), cryptoCGA (780 in V, 658 in O), neoCGA (473 in V, 446 in O), xylitol (481 in V), and α-tocopherol (283 in V). Phenolic compounds were detected with higher contents in V and O varieties as opposed to B: CGA (V/O 112/79 times higher than B), cryptoCGA (V/O 130/110 times higher than B), neoCGA (V/O 100/94 times higher than B), α-tocopherol (V/O >2000/>2 times higher than B), and rutin (V/O 28/26 times higher than B).

Table 3 shows the compound distribution depending on the microfractionation extraction polarity. Compounds **1**–**9** in Table 2 and Table 3 tended to distribute in both the nonpolar and polar extracts. The polar compounds inclusive of phenolic compounds, polyhydric alcohols, and purine derivative (compounds **10**–**18** in Table 2 and Table 3) were only detectable in polar extracts. Compounds with nonpolar structures inclusive of terpenoids and long chains (compounds **19**–**24** in Table 2 and Table 3) tended to only be detected (if at all) in the nonpolar extracts.

The inherent separative capabilities of Bifrac N™ microfractionation technology allowed us to simultaneously and quantitatively extract and recover multiple compounds from a highly decomplexified sample. The recovered content range of 0.1–2277 µg/g dry material enables the detection of both minor (e.g., myosmine) and highly abundant (e.g., rutin) components from the same crude extract, providing a high-resolution sample preparation approach that can compare the potencies for the biosynthesis of different secondary metabolites of interest. 

### 2.3. Single Solvent Extraction Optimization

Considering the above results, the goal was to optimize the extraction efficiency for the compounds of interest, with the previously defined more favorable extraction solvents based on the data obtained from rapid solubility tests and microfractionation. To fine tune the solvent combination, we decided to test seven approaches: (1) CH_2_Cl_2_, (2) CH_2_Cl_2_/MeOH (4:1, *v*/*v*), (3) CH_2_Cl_2_/MeOH (1:1, *v*/*v*), (4) CH_2_Cl_2_/MeOH (1:4, *v*/*v*), (5) MeOH, (6) MeOH/H_2_O (7:3, *v*/*v*), and (7) MeOH/H_2_O (9:1, *v*/*v*).

Three different base extraction (BM) methods (BM1, BM2, and BM3) were also investigated (Table 4 and Appendix A). BM3, which is more of a classical method, uses a higher solvent volume to dry plant material ratio and longer extraction time than BM1 and BM2, while the latter two use an ultrasonic approach, as detailed in the Section 4. Method optimization was only conducted on the *N. rustica* variety Bakoum Miena in order to objectively compare only the effects of the method and solvent tested and exclude differences in the matrix effect, which were assessed in Section 2.3.

The concentration levels of bioactive compounds were monitored using various extraction method conditions (Table 4). All alkaloids (nicotine, anatabine, anabasine, myosmine, nornicotine) including Vitamin B_3_ (nicotinic acid) and nicotinamide were best extracted with MeOH/H_2_O (7:3, *v*/*v*, M13), except for cotinine and norcotinine that were better extracted using CH_2_Cl_2_ as the solvent and BM2. The chlorogenic acid isomers (CGA, neoCGA, and cryptoCGA) were better extracted using MeOH/H_2_O (9:1, *v*/*v*) and BM2. The same best extraction solvent method was obtained for *trans*-zeatin, rutin, and xylitol, but similar extraction yields were obtained with BM1 or BM2.

While we did not find a ubiquitously applicable extraction method, we were able to identity a preferred extraction solvent and confirm the importance of sonication and extraction time for all these compounds of interest. This information can serve as a steppingstone for finetuning an ideal extraction for unique compounds of interest in the future, especially for the simultaneous extraction of several compounds. 

### 2.4. Profiling of Six Nicotiana Varieties for Potentially Valuable Compounds

It is equally important to determine the plant variety most suited for the production and extraction of our compounds of interest. To objectively benchmark the concentration levels of bioactive compounds, six different *Nicotiana* varieties were studied using a single solvent combination and extraction method. Although the single solvent extraction method is not optimal to generate the highest yields for all targeted compounds, it provides a rational comparison approach of the bioactive compound content produced by the different plant varieties. Thus, MeOH/H_2_O (7:3, *v*/*v*) was chosen as 7 of the 15 bioactive compounds (Table 4) showing the best yield results under this extraction condition. Table 5 summarizes the concentration levels obtained across the six *Nicotiana* varieties: (1) Virginia, (2) Burley, (3) Oriental, (4) NRT61, (5) NRT63, and (6) Bakoum Miena. 

Removal of water content (i.e., drying) of plant crude material as well as removal of (organic) solvents under vacuum and/or heating (even only at 40 °C) can result in significant loss of volatile compounds. To avoid the potential loss of bioactive compounds of interest, although leaves were cured, we made a point to not perform any additional drying steps during sample preparation prior to LC-HR-MS quantification.

Rutin and CGA had the best extraction yields from the Virginia variety, reaching up to 225 and 119 mg/g dry plant material, respectively. Compared to Burley, these yields were 28.4- and 91-fold higher in Virginia. For the major tobacco alkaloids, nicotine and anabasine contents were highest in Bakoum Miena (5.2-fold higher compared to the Oriental variety), while anatabine was highest in Virginia and Burley varieties (up to 5.9-fold compared to Oriental). Myosmine was almost equally extracted from all varieties except Virginia, which had only half the content. Cotinine was highest in Burley and Oriental, nornicotine was fairly high all over but was more available in Oriental, norcotinine and nicotinic acid were found more in Burley, and FA was preferentially extracted from Virginia. LA showed much higher recovery from NRT63. The majority of these results are well in line with recently published work where a high number of *Nicotiana* genus species were assayed for their alkaloid content [10].

Although the absolute compound recovery values for the single solvent profiling experiment are not directly comparable to the ones referring to the microfractionation coverage in the *N. tabacum* varieties (Table 2), the trends are similar for most of the compounds. For those compounds where the trends are not similar, the discrepancy could be explained by the inherent difference in the extraction methods (volumes of solvent used, starting material weight, temperature etc.) and technologies applied.

### 2.5. Projected Yield of Potentially Valuable Compounds in Six Nicotiana Varieties

Table 6 shows the theoretical yields of valuable tobacco compounds as expressed in kilograms per hectare. This is particularly informative when considering industrial extraction related to the demand of sustainable production of non-fossil fuel-based building blocks as potential starting material for establishing bioeconomy value-chains in the pharmaceutical industry [101]. Compounds need to reach a specific percentage of plant biomass dry weight to make their extraction economical; the acceptable concentration limit is highly dependent on compound market values. The theoretical yield table also helps compare different species and varieties for their biosynthetic potential. 

## 3. Discussion

Using the tobacco plant as a “bioreactor” makes economic sense, especially because it is not used as a food crop. It presents an excellent opportunity for sourcing bioactive compounds without affecting the critical issue of food production. Containing 90,000 genes and 5700 unique and already described chemicals, the tobacco plant has high potency and versatility for producing numerous compounds [3,4]. Accessing this diverse set of compounds will require a practical approach to isolate them from the complex plant matrix.

In the present work, we propose how to choose from multiple extraction and purification workflows depending on the goals. On one hand, when purifying a single family of compounds (e.g., alkaloids), one can adopt the optimized single extraction solvent MeOH/H_2_O (7:3 *v*/*v*) and perform method M13 to maximize the recovery. On the other hand, our results show how it is possible to optimize a fractionation method to simultaneously recover multiple compound families from the same plant crude extract, thus minimizing waste and maximizing rentability. In the first case, the results provide details of extraction (solvent, method, etc.) and facilitate the choice of starting plant variety both from the point of view of target compound quantity (Table 5) and the total anticipated material produced from the field of the variety in question (Table 6). To give an example: if anabasine is our compound of interest, although its availability using our extraction method was maximal in the Bakoum Miena variety (833 µg/g dry plant material), because Burley has a five-fold total dry mass yield per hectare, the latter variety should be chosen to achieve a higher final yield equivalent of 5.13 × 242 µg/g = 1242 µg/g dry plant material.

One aspect that was not pursued in this work but could be further studied is volatile compound extraction. While some of the compounds of interest are enriched by curing/drying leaf material, others such as terpenoids (e.g., farnesol, linalool, caryophyllene, borneol, α-pinene, and limonene) were probably decreased by the grinding, homogenization, and drying steps of the extraction process. Extracting these volatile compounds should be performed from fresh leaf material and supercritical fluid extraction procedures (not evaluated in the manuscript).

We should keep in mind that the final goal is to extract and purify single bioactive compounds or simple mixtures (natural product extracts) with determined bioactivities. Single solvent extraction is the first enrichment step that needs to be followed by fractionation (chromatographic or otherwise) to achieve expected purification levels of the compounds of interest. 

Another potential application that this work lays the ground for is bioactivity-guided fractionation [102]. As described in the Introduction, numerous valuable compounds can be sourced from the *Nicotiana* species. Many of them have potential therapeutic applications such as anti-inflammatory effects (e.g., rutin, anatabine, FA, CGA). As a future approach, bioassay-guided identification of bioactive fractions and compounds within these fractions can be pursued.

## 4. Materials and Methods

### 4.1. Plant Sample Generation

Leaf samples of three *N. rustica* tobacco varieties (Bakoum Miena, NRT63, and NRT61) and two *N. tabacum* varieties (Burley and Virginia) were cultivated in 2021 in Italy as provided by PMI (Italy/Switzerland). The Oriental variety was grown and provided in 2021 by PMI-associated farmers located in Turkey (PMI Turkey). All plants were cultivated by following common agricultural practices. Samples were dried for 96 h at 40 °C. Cured and dried samples were milled using a Retsch rotary millrotor beater mill SR 300 (Retsch GmbH, Haan, Germany) to a particle size of less than 200 µm. The milled samples were then stored at room temperature until they were utilized.

### 4.2. Reagents and Chemicals

Nicotine, anabasine, myosmine, nicotinamide, cotinine, nornicotine, norcotinine, nicotinic acid, and their corresponding deuterated or ^13^C internal standards were purchased from Sigma-Aldrich (St. Louis, MO, USA) unless otherwise stated below. Anatabine was purchased from Cayman Chemicals (Ann Arbor, MI, USA), and anatabine-d4 and myosmine-d4 were obtained from Toronto Research Chemicals (Ontario, ON, Canada). Solvents used for LC-MS analysis were from Honeywell (Charlotte, NC, USA). Ammonium acetate was from Sigma-Aldrich, and formic acid was from Thermo Fisher Scientific (Waltham, MA, USA).

The following solvents used for microfractionation were purchased from Shanghai Titan Scientific Co. Ltd. (Shanghai, China): butanol (≥99.5%, certified analytical reagent (AR)), dichloromethane (≥99.5%, AR), ethyl acetate (≥99.5%, AR), 95% ethanol (95%, AR), ethanol (≥99.7%, AR), hexane (≥97.0%, AR), and methanol (≥99.5%, AR). Ultrapure water was produced by Bicoll (Planegg, Germany). The pipettes for 96-well microtiter plates were produced by Eppendorf (Hamburg, Germany). Polytetrafuoroethylene membrane filters (0.22 µm) and syringes were purchased from Anpel technologies (Shanghai, China).

### 4.3. Rapid Solubility Test of Potentially Valuable Compounds

For each of the 29 reference compounds, 2 mg was weighed into a 15 mL glass tube and dissolved in the minimum amount of the test solvent by gradually adding the solvent with a micropipette until complete dissolution or to a final volume of 10.0 mL. Seven solvents with polarities from nonpolar to polar were chosen for the test: hexane (C_6_H_14_), CH_2_Cl_2_, CH_2_Cl_2_/MeOH (4:1, *v*/*v*), EtOH, MeOH, MeOH/H_2_O (7:3, *v*/*v*), and H_2_O. The consumed solvents and their volumes were recorded for each reference compound.

### 4.4. Microfractionation 

Dried and powdered plant material of the three *N. tabacum* varieties (Virginia, Burley, and Oriental) were extracted with the mixed solvent of CH_2_Cl_2_ and ethyl acetate (EtOAc) for nonpolar fractions, and 95% EtOH for polar fractions successively following the standard protocols at Bicoll, yielding a nonpolar crude extract and a polar crude extract for each variety.

The nonpolar crude extracts of the same three *N. tabacum* varieties were fractionated with the mixtures of hexane (C_6_H_14_) and EtOAc and MeOH and H_2_O, and the polar crude extracts of the three varieties were fractionated with the mixtures of EtOAc and butanol and H_2_O following the standard protocols at Bicoll, yielding to the 96 nonpolar microfractions and 96 polar microfractions per variety. These 192 microfractions (≥0.5 mg per well) were equalized in sample weight (0.2 mg per well) by plating in a 96-well microtiter plate, which was further used for quantification measurements by LC-HR-MS.

### 4.5. Extraction Method Optimization

BM1—Base method 1:

Powdered plant material (0.5 g) was weighed into a 50 mL centrifuge tube, supplemented with 15 mL of extraction solvents, and extracted in an ultrasonic bath for 20 min at room temperature. Then, 0.25 g active carbon (75 µm) and 0.13 g PSA (40–60 µm; [3-(2-Aminoethylamino)propyl]trimethoxysilane) were added and intensively agitated for 1 min and then centrifuged at 10,000 rpm for 5 min. The supernatant was filtered through a 0.22 µm syringe membrane filter, and 1 mL filtrate was blow dried with argon at room temperature. Finally, the dried sample was dissolved in 1 mL of MeOH, filtered through a 0.22 µm membrane before LC-HR-MS quantification.

BM2—Base method 2:

Powdered plant material (0.5 g) was weighed into a 50 mL centrifuge tube, supplemented with 15 mL of extraction solvents, and extracted in an ultrasonic bath for 20 min at room temperature, before centrifugation at 10,000 rpm for 5 min. The supernatant was filtered through a 0.22 µm syringe membrane filter, and 1 mL filtrate was blow dried with argon at room temperature. Finally, the dried sample was dissolved in 1 mL of MeOH and filtered through a 0.22 µm membrane before LC-HR-MS quantification.

BM3—Base method 3:

Powdered plant material (0.2 g) was weighed into a 50 mL centrifuge tube, supplemented with 20 mL of MeOH/H_2_O (7:3), placed on a rotary shaker at room temperature for 72 h at 50 rpm, and then centrifuged at 10,000 rpm for 5 min. The supernatant was filtered through a 0.22 µm syringe membrane filter, and 1 mL filtrate was blow dried with argon at room temperature. Finally, the dried sample was dissolved in 1 mL of MeOH and filtered through a 0.22 µm membrane before LC-HR-MS quantification.

### 4.6. Solvent Extraction from Six Tobacco Varieties

Collected plant material was lyophilized and pulverized by shaking at 400 rpm for 8 h in containers with glass beads. Samples for LC-MS analyses were prepared by extracting approximately 25 mg of fine powder with 5 mL of the appropriate extraction solvent by agitating on a rotary shaker for 24 h at room temperature, then filtered with a Sterile PES Syringe Filter with pore size of 0.2 µm (Fisherbrand), and diluted 1:20 for injection into the LC-HR-MS system without a drying step post extraction. 

### 4.7. Potentially Valuable Compound Quantification by LC-HR-MS

List of analytes and internal standards and their selected properties: ESI polarity, monoisotopic mass (*m*/*z*), and retention time (RT) are represented in Appendix A.

#### 4.7.1. LC-HR-MS Method 1

Simultaneous determination of nine alkaloids (nicotine, anatabine, anabasine, myosmine, nicotinamide, cotinine, nornicotine, norcotinine, and nicotinic acid) and their respective deuterated or ^13^C internal standards (IS) was performed on a Vanquish Duo UHPLC system coupled to an Orbitrap IDX mass spectrometer (Thermo Fisher Scientific) operating in positive electrospray ionization mode scanning the 50–800 amu mass range at 60k resolution. Chromatographic separation of the injected sample (3 µL) was achieved on an Acquity HSS T3 column, (150 × 2.1 mm, 1.7 µm from Waters, Milford, MA); the column temperature was set to 45 °C. Ammonium acetate in water (10 mM; pH 8.9; mobile phase A) and ammonium acetate in methanol (10 mM; mobile phase B) were applied as a gradient (0–0.25 min: 10% B; 4.25 min: 98% B; 5.5 min: 98% B, 5.6 min: 10% B until 7 min) with a constant flow rate of 0.3 mL/min. 

#### 4.7.2. LC-HR-MS Method 2

Chromatographic separation of analytes (CGA, cryptoCGA, neoCGA, rutin, and *cis*- and *trans*-zeatin) and deuterated internal standards was performed on an Xbridge PHENYL column (100 × 2.1 mm, 2.5 µm particle size from Waters). A gradient of mobile phase A: 0.1% formic acid in water and mobile phase B: 0.1% formic acid in acetonitrile was applied with the following schedule (0–0.5 min: 5% B; 8 min: 98% B; 8.5 min: 98% B, 8.51 min: 5% B until 10 min) with a constant flow rate of 0.4 mL/min. The 10 µL samples solubilized in 5% acetonitrile in water were injected and analyzed in positive ionization mode scanning the 80–800 amu mass range at 60k resolution on the same instrument as described above. Column temperature was set to 40 °C.

#### 4.7.3. LC-HR-MS Method 3

Chromatographic separation of analytes ambroxide, LA, α-tocopherol, Vitamin D_3_, and ^13^C or deuterated internal standards was performed on an Xbridge PHENYL column (100 × 2.1 mm, 2.5 µm particle size from Waters). A gradient of mobile phase A: 0.1% formic acid in water and mobile phase B: 0.1% formic acid in acetonitrile was applied with the following schedule (0–0.5 min: 50% B; 8 min: 98% B; 8.5 min: 98% B, 8.51 min: 50% B until 10 min) with a constant flow rate of 0.4 mL/min. The 10 µL samples solubilized in 5% acetonitrile in water were injected and analyzed in positive ionization mode scanning the 80–800 amu mass range at 60k resolution on the same instrument as described above. Column temperature was set to 40 °C.

#### 4.7.4. LC-HR-MS Method 4

Xylitol and its deuterated internal standard were separated on a Cosmosil Sugar D column (150 × 2.1 mm) (Nacalai Tesque, Kyoto, Japan). A gradient of mobile phase A: water and mobile phase B: acetonitrile was applied with the following schedule (0–0.5 min–90% B; 1.2 min: 65% B; 3 min: 50% B, 5.5 min: 50% B, 5.6 min: 90% B until 10 min) with a constant flow rate of 0.4 mL/min. Then, 10 µL sample solubilized in 50% acetonitrile in water was injected and analyzed in negative ionization mode scanning 80–800 amu mass range and 60k resolution; column temperature was set to 40 °C.

## 5. Conclusions

Secondary metabolites of plants are a cornerstone for the bio-economy transformation of industrial processes, both as lead compounds as well as final products. Non-food plants such as tobacco offer further options for optimization through classical crop breeding or state-of-the art, tailor-made approaches modifying the genetic make-up of the plants with the purpose of using them as a “bioreactor” to produce high-value compounds of interest. As the available worldwide CO_2_ budget requirement for coping with climate anomalies becomes more stringent every year, this work promotes relevant transformation processes for diversifying industrial feedstocks. 

Herein, we described the fractionation and extraction optimization of several potentially valuable compounds and their profiles in six different varieties from two *Nicotiana* species that could be of economic interest in the future. The obtained results provide essential basic data that inform the initial choice of tobacco plant species/varieties and extraction conditions when designing efficient processes for the production of maximum amounts of single compounds of interest, or for the most suitable combination of extracted products in a biorefinery approach.

## Figures and Tables

**Figure 1 molecules-27-08105-f001:**
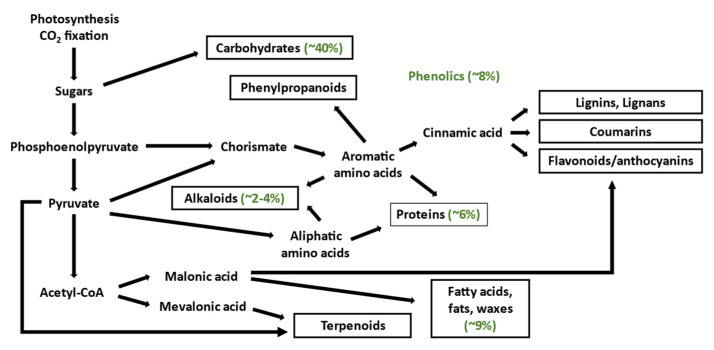
Tobacco biosynthetic pathways leading to primary and secondary metabolites. The green text shows the approximate dry weight composition of the most important compounds [5]. Desired classes of commercially viable compounds are indicated in black rectangles. This figure was assembled from several different scientific sources [5,6,7,8].

**Table 1 molecules-27-08105-t001:** Rapid solubility test results of 29 bioactive compounds in 7 different solvents (mg/mL, at room temperature (21–23 °C).

No.	Compounds	C_6_H_14_	CH_2_Cl_2_	CH_2_Cl_2_/ MeOH (4:1)	EtOH	MeOH	MeOH/ H_2_O (7:3)	H_2_O
1	Anatabine	217	442	**512**	490	444	193	100
2	Anabasine	NS	10.3	70.0	97.0	103	**109**	NS
3	Myosmine	4.27	194	175	**418**	356	231	180
4	Nicotinamide	NS	NS	37.3	20.8	66.3	114	**150**
5	Nicotinic acid	NS	NS	3.98	3.51	**6.71**	2.85	3.10
6	Cotinine	NS	203	208	190	**235**	139	145
7	Nornicotine	358	422	412	396	**428**	418	416
8	Zeatin	NS	NS	0.61	0.56	2.28	**3.03**	NS
9	Rutin	NS	NS	3.82	6.67	**49.0**	0.52	NS
10	Chlorogenic acid	NS	NS	1.82	23.3	**31.7**	1.27	0.24
11	Ferulic acid	NS	NS	30.3	33.8	**51.5**	10.5	NS
12	Solanesol	42.0	**235**	203	3.31	0.51	NS	NS
13	Zeaxanthin	NS	0.29	**0.49**	NS	NS	NS	NS
14	Lutein	NS	NS	**4.66**	NS	NS	NS	NS
15	Linoleic acid	374	504	502	**558**	526	NS	NS
16	Sclareol	3.76	58.0	**219**	75.7	65.7	3.17	NS
17	Ambroxide	66.0	**130**	116	51.8	49.8	NS	NS
18	Vitamin D_3_	104	100	**212**	74.3	113	NS	NS
19	Vitamin E	215	208	**241**	43.0	31.2	0.23	NS
20	Squalene	446	**474**	472	5.94	0.22	NS	NS
21	Phytol	**470**	432	416	430	408	0.32	NS
22	Farnesol	400	382	348	376	**466**	0.75	NS
23	Neophytadiene	**486**	**486**	482	57.5	4.08	NS	NS
24	Linalool	422	440	**462**	454	**462**	77.3	0.57
25	Caryophyllene	376	382	**480**	196	36.2	NS	NS
26	Megastigmatrienone	1.1	215	212	197	**228**	38.8	NS
27	Borneol	29.9	80.7	202	218	**504**	71.7	NS
28	α-Pinene	388	428	**432**	406	71.0	NS	NS
29	Limonene	**442**	398	388	392	394	NS	NS

NS: not soluble, insoluble in the solvent at the tested concentration range (0.19–558 mg/mL). Number in bold: the min. solubility of the compounds in the best solvent.

**Table 2 molecules-27-08105-t002:** Total determined contents of 22 reference compounds in the three varieties: Virginia, Burley, and Oriental (µg/g dry plant material).

No.	Compounds	Virginia	Burley	Oriental
1	Nicotine	1245	683	379
2	Anatabine	104	74.6	18.9
3	Anabasine	18.6	14.1	4.39
4	Myosmine	0.30	0.11	0.29
5	Nicotinamide	0.22	0.21	0.80
6	Cotinine	2.93	4.99	13.2
7	Nornicotine	34.4	37.0	63.5
8	Norcotinine	0.68	0.99	1.34
9	Nicotinic acid	5.16	6.73	6.41
10	Chlorogenic acid	1574	14.0	1109
11	Cryptochlorogenic acid	780	5.99	658
12	Neochlorogenic acid	473	4.72	446
13	*cis*-Zeatin	0.11	0.16	0.19
14	*trans*-Zeatin	2.04	3.37	0.97
15	Ferulic acid	N/F	N/F	0.10
16	Isoferulic acid	4.89	0.94	2.15
17	Rutin	2277	82.4	2129
18	Xylitol	481	37.1	110
19	Ambroxide	N/F	0.03	0.10
20	Linoleic acid	85.8	2.07	N/F
21	α-Tocopherol	283	N/F	0.21
22	Vitamin D_3_	N/F	N/F	N/F

**Table 3 molecules-27-08105-t003:** Total determined contents of 22 compounds of interest in the nonpolar and polar fractions of the three varieties: Virginia, Burley, and Oriental (µg/g dry plant material).

No.	Compounds	Virginia		Burley		Oriental	
		Nonpolar	Polar	Nonpolar	Polar	Nonpolar	Polar
1	Nicotine	196	1049	109	574	105	274
2	Anatabine	8.38	95.4	10.2	64.5	3.21	15.7
3	Anabasine	0.92	17.7	1.18	12.9	0.70	3.69
4	Myosmine	0.04	0.26	0.02	0.09	0.10	0.19
5	Nicotinamide	0.12	0.09	0.05	0.16	0.52	0.28
6	Cotinine	2.31	0.63	3.52	1.47	12.2	0.92
7	Nornicotine	1.85	32.5	1.99	35.0	4.44	59.0
8	Norcotinine	0.16	0.52	0.33	0.66	0.88	0.46
9	Nicotinic acid	2.88	2.28	2.21	4.52	4.46	1.95
10	Chlorogenic acid	N/F	1574	N/F	14.0	N/F	1109
11	Cryptochlorogenic acid	N/F	780	N/F	5.99	N/F	658
12	Neochlorogenic acid	N/F	473	N/F	4.72	N/F	446
13	*cis*-Zeatin	N/F	0.11	N/F	0.16	N/F	0.19
14	trans-Zeatin	N/F	2.04	N/F	3.37	N/F	0.97
15	Ferulic acid	N/F	N/F	N/F	N/F	N/F	0.10
16	Isoferulic acid	N/F	4.89	N/F	0.94	N/F	2.15
17	Rutin	N/F	2277	N/F	82.4	N/F	2129
18	Xylitol	N/F	481	N/F	37.1	N/F	110
19	Ambroxide	N/F	N/F	0.03	N/F	0.10	N/F
20	Linoleic acid	85.8	N/F	2.07	N/F	N/F	N/F
21	α-Tocopherol	283	N/F	N/F	N/F	0.21	N/F
22	Vitamin D_3_	N/F	N/F	N/F	N/F	N/F	N/F

N/F: not found.

**Table 4 molecules-27-08105-t004:** Selected compound quantification from Bakoum Miena sample extracts (µg/g dry plant material).

Method		M1	M2	M3	M4	M5	M6	M7	M8	M9	M10	M11	M12	M13
Base Method		BM1	BM2	BM1	BM2	BM1	BM2	BM1	BM2	BM1	BM2	BM1	BM2	BM3
	Solvent	CH_2_Cl_2_		CH_2_Cl_2_/MeOH (4:1, *v*/*v*)		CH_2_Cl_2_/MeOH (1:1, *v*/*v*)		CH_2_Cl_2_/MeOH (1:4, *v*/*v*)		MeOH		MeOH/H_2_O (9:1, *v*/*v*)		MeOH/H_2_O (7:3, *v*/*v*)
No.	Compound													
1	Nicotine	8416	7093	22,432	24,480	29,515	31,401	33,803	28,978	33,228	38,819	36,101	36,775	**47,827**
2	Anatabine	18	19	232	261	270	319	264	332	231	357	283	369	**450**
3	Anabasine	20	24	274	311	332	364	327	385	289	408	354	408	**542**
4	Myosmine	5.28	5.04	8.50	7.19	8.34	6.04	6.80	6.26	4.15	6.63	4.42	6.67	**13.82**
5	Nicotinamide	0.69	1.16	2.73	2.80	2.84	3.09	2.74	3.11	2.33	3.46	2.65	3.07	**4.09**
6	Cotinine	135	**231**	159	152	70	141	44	133	38	139	40	50	72
7	Nornicotine	36	46	204	303	251	337	220	394	207	368	311	396	**538**
8	Norcotinine	5.08	**8.58**	4.58	7.53	4.60	3.95	4.12	4.07	3.57	4.27	3.89	2.87	3.92
9	Nicotinic acid	25.9	40.9	61.1	76.3	52.1	78.9	52.5	71.6	46.8	78.3	58.9	66.9	**96.2**
10	cryptoCGA	N/F	N/F	1253	5208	7957	21,192	11,518	29,854	9603	39,877	21,470	**48,775**	17,102
11	neoCGA	N/F	N/F	1193	5282	9017	25,057	13,972	38,252	10,746	52,038	23,618	**61,684**	16,901
12	CGA	N/F	N/F	4601	16,421	19,338	44,408	25,938	56,836	23,262	73,164	42,292	**85,533**	38,199
13	*trans*-Zeatin	2.6	3.5	72.0	88.6	92.5	100.9	105.2	109.2	105.2	121.9	131.6	**134.6**	43.0
14	Rutin	7	6	946	23,554	13,493	97,419	13,457	12,3712	8163	156,806	10,947	**185,959**	65,277
15	Xylitol	62	61	3474	3212	7260	6545	9950	8250	10,643	10,950	**14,453**	14,001	4938

The highest recovery concentrations are shown in bold.

**Table 5 molecules-27-08105-t005:** Selected compound profiling in different *N. tabacum* (Virginia, Burley, Oriental) and *N. rustica* (NRT61, NRT63, Bakoum Miena (B.M.)) varieties. (data are expressed as µg/g dry plant material).

No.	Compound	Virginia	Burley	Oriental	NRT61	NRT63	B.M.
1	**Nicotine**	34,609	37,162	18,055	76,696	62,926	**94,150**
2	**Anatabine**	**1699**	**1544**	287	663	606	622
3	**Anabasine**	281	242	61	738	733	**833**
4	**Myosmine**	6.69	**13.13**	**13.24**	11.00	11.29	**13.23**
5	**Nicotinamide**	2.50	1.65	**10.61**	4.87	4.34	5.36
6	Cotinine	34.8	**116.4**	**103.4**	59.8	69.8	46.3
7	**Nornicotine**	803	775	**1138**	**967**	**962**	909
8	Norcotinine	3.01	**7.19**	3.93	4.21	4.76	3.26
9	**Nicotinic acid**	87.3	**179.8**	55.7	68.8	77.1	97.4
10	CGA	**118,846**	1305	69,838	39,589	28,146	26,955
11	Rutin	**224,699**	7910	114,419	139,420	118,015	122,224
12	*Ferulic acid*	**309**	63	203	69	70	53
13	*Linoleic acid*	6496	2181	5821	2979	**26,023**	2967

Compounds for which the solvent extraction method involving MeOH/H_2_O (7:3, *v*/*v*) was the optimal extraction method are shown in bold. Compounds in italic, although not optimized for extraction efficiency because of their potential value, were included in the tobacco variety profiling experiments.

**Table 6 molecules-27-08105-t006:** Theoretical yield in kilograms per hectare (kg/ha) of valuable tobacco compounds based on recorded and typical mass yields of tobacco species and varieties from our internal database.

No.	Compound	Virginia	Burley	Oriental	NRT61	NRT63	B.M.
	Total Dry Weight	3288.91	5222.19	1700	2238.95	1433.6	1017.92
1	Nicotine	113.83	194.07	30.69	171.72	90.21	95.84
2	Anatabine	5.59	8.06	0.49	1.48	0.87	0.63
3	Anabasine	0.92	1.26	0.10	1.65	1.05	0.85
4	Myosmine	0.02	0.07	0.02	0.02	0.02	0.01
5	Nicotinamide	0.01	0.01	0.02	0.01	0.01	0.01
6	Cotinine	0.11	0.61	0.18	0.13	0.10	0.05
7	Nornicotine	2.64	4.05	1.93	2.17	1.38	0.93
8	Norcotinine	0.01	0.04	0.01	0.01	0.01	0.00
9	Nicotinic acid	0.29	0.94	0.09	0.15	0.11	0.10
10	CGA	390.87	6.81	118.72	88.64	40.35	27.44
11	Rutin	739.01	41.31	194.51	312.15	169.19	124.41
12	Ferulic acid	1.02	0.33	0.35	0.15	0.10	0.05
13	Linoleic acid	21.36	11.39	9.90	6.67	37.31	3.02

## Data Availability

Not applicable.

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
