# Peer review of "Fractionation and Extraction Optimization of Potentially Valuable Compounds and Their Profiling in Six Varieties of Two Nicotiana Species"

_molecules, 2022, doi:10.3390/molecules27228105_

Round 1

Reviewer 1 Report

Line 24. Why were three varieties of tobacco analyzed instead of six? It does not match the content of the article

Line 326. Determination of the small molecule content in a complex plant matrix depends on the understanding of minimum solubility in the chosen extraction solvent. How did you get this view?

Line 353. Solvent polarity is independent of pH. How did you get the importance of plant matrices in terms of pH associated to the plant starting material ?

Line 418. The format of the H2O is wrong.

Line 425. The table does not match the content of the article. The extraction solvent used in method M13 of Table S1 is not in the approaches.

Line 426. Base Extraction methods BM03 is only used for method M13, not for further study. For example, using the extraction solvent MeOH/H2O (9:1, v/v) and method BM03, can the extraction result be better? 

Author Response

We would like to sincerely thank the Reviewer for helping us improve the manuscript to better represent the scientific data we collected in our study and for identifying our mistakes, making it more accessible for the intended audience.

Q: Line 24. Why were three varieties of tobacco analyzed instead of six? It does not match the content of the article

 A: We actually analyzed 6 varieties in total, 3 from N tabacum and 3 from N rustica. Because indeed the text was not clear on this point we changed the original sentence : “We proceeded with profiling some of these compounds in three varieties from both Nicotiana tabacum and Nicotiana rustica species, identifying the optimal variety for each.” to “We proceeded with profiling some of these compounds in a total of six varieties from Nicotiana tabacum and Nicotiana rustica species, identifying the optimal variety for each.”

Q: Line 326. Determination of the small molecule content in a complex plant matrix depends on the understanding of minimum solubility in the chosen extraction solvent. How did you get this view?

 A: If the target compound (small molecule) has a good or OK/sufficient solubility in the chosen extraction solvent, it tends to be solved well during the extraction process and therefore completely extracted and measured, and vise versa. Taking the experimental data of chlorogenic acid (CGA) and rutin in Table 1 and 4 for explaination:

  • CGA showed no solubility (NS) in the solubility test in CH2Cl2 and an OK solubility (31.7 mg/mL) in MeOH (Table 1), and in the content determination in the plant Bakoum Miena sample extracts, it showed, correspondingly, NF (Not Found) in CH2Cl2 extracts with both BM01 and BM02 extraction methods while 23,262 (BM01) and 73,164 (BM02) µg/g dry plant material in MeOH extracts.
  • Rutin showed no solubility (NS) in the solubility test in CH2Cl2 and an OK solubility (49.0 mg/mL) in MeOH (Table 1), and in the content determination in the plant Bakoum Miena sample extracts, it showed, correspondingly, 7 (BM01) and 6 (BM02) µg/g dry plant material in CH2Cl2 extracts with both BM01 and BM02 extraction methods while 8,163 (BM01) and 156,806 (BM02) µg/g dry plant material in MeOH extracts.

That means, although there is a considerable content of CGA and rutin in the plant material, the wrong choice of extraction solvent (CH2Cl2) will lead to a totally misleading result (NF), and only the right choice of extraction solvent (MeOH or similar) will have a chance to get close to the content produced by the plant and the theoretical maximum possible to be extracted material. Therefore the understanding of minimum solubility of chosen extraction solvent are becoming mandatory in the content determination experiments and disclosure of results in this publication.

Q: Line 353. Solvent polarity is independent of pH. How did you get the importance of plant matrices in terms of pH associated to the plant starting material ?

A: Solvent polarity is independent of pH, but the plant matrices might show different pH or ionization properties in different solvents.

Taking CGA and rutin as example:

  • Both two compounds are best soluble in MeOH (49.0 mg/mL for rutin, 31.7 mg/mL for CGA) and poorly soluble in MeOH/water (7:3) – 0.52 mg/mL for rutin, 1.27 mg/mL for CGA (see Table 1)
  • The highest content (corresponding to best extraction solvent, see Table 4) are not MeOH (156,806 for rutin, 73,164 for CGA µg/g dry plant material with M02), but MeOH/water (9:1) - 185,959 for rutin, 85,533 for CGA.
  • Interpretation: rutin and CGA are phenolic compounds with acidic properties. As pure organic compounds, they are solved better in pure MeOH than neutral MeOH-water system, but will be solved better in basic MeOH-water system. Here the plant matrices Bakoum Miena contains abundant alkaloids which gets easier ionization in MeOH-water system (becoming more basic – lower pH) than pure MeOH.
  • The same story for cryptoCGA and neoCGA.

Therefore if pH is associated to plant starting material, the solubility of some compounds will be quite different and demonstrating varied content in the determination results.

Q: Line 418. The format of the H2O is wrong.

A: Formatting has been changed.

Q:  Line 425. The table does not match the content of the article. The extraction solvent used in method M13 of Table S1 is not in the approaches.

A: The typo was corrected from MeOH/H2O (7:1, v/v)  to MeOH/H2O (7:3, v/v) in Table S1 that it now matches the approaches described in lines 421-423.

Q: Line 426. Base Extraction methods BM03 is only used for method M13, not for further study. For example, using the extraction solvent MeOH/H2O (9:1, v/v) and method BM03, can the extraction result be better? 

Indeed BM3 was only used for M13 as this method was considered as the benchmark method from our previous experience and from different publications.

Reviewer 2 Report

The proposed paper "Fractionation and Extraction Optimization of Potentially Valuable Compounds and Their Profiling in Six Varieties of Two Nicotiana Species" presents some interesting results of optimization of extraction and fractionation of the secondary metabolites in different varieties of Nicotiana species. The results obtained in the study can be a valuable contribution to the field and can not only be used for scientific research but also by the industry.
 The paper is well-written and relatively well structured; however, in my opinion, it would be useful to add some additional information.
Please, find some comments below:

 Please provide more detailed information on the drying conditions of the plant material.
 If possible, please briefly describe the standard Bicoll protocols used in the study (4.4 Microfractionation)
 Also, information on the GC-HR-MS method used to quantify the compounds for the microfractionation assessment is missing in the section: "Materials and methods".
Should other conditions than those mentioned for LC-HR-MS be used, it would also make sense to indicate them.

There is an inconsistency in the presentation of base extraction methods: starting from lines: 432, the authors use BM2, BM1, also in Table 4. Please correct this, or if these are methods other than BM01... please give a description.
Small note: It would be good to discuss the results in the discussion section instead of the results section.

The conclusions section looks more like an introduction or an abstract. Please, rewrite and include the outcomes of the study.

Author Response

We would like to sincerely thank the Reviewer for helping us improve the manuscript to better represent the scientific data we collected in our study and for identifying our mistakes, making it more accessible for the intended audience.

Q: Please provide more detailed information on the drying conditions of the plant material.

A: Samples were dried for 96h at 400C. Included in line 549.

Q: If possible, please briefly describe the standard Bicoll protocols used in the study (4.4 Microfractionation)

A: Microfraction is a readily available approach (see. ref 97B) and it is mentioned, like the commercially available material in material and methods.

 Q: Also, information on the GC-HR-MS method used to quantify the compounds for the microfractionation assessment is missing in the section: "Materials and methods".
Should other conditions than those mentioned for LC-HR-MS be used, it would also make sense to indicate them.
 A: Although in the beginning of the study we did have compounds that were measured by GC-MS these were removed due to inconsistency from all results. All mention of”GC-MS” should have also been removed. We rectify our mistake by removing the following words from the manuscript lines 381-382: “and gas chromatography high resolution mass spectrometry (GC-HR-MS)”

Q: There is an inconsistency in the presentation of base extraction methods: starting from lines: 432, the authors use BM2, BM1, also in Table 4. Please correct this, or if these are methods other than BM01... please give a description.

A: The base method codes were corrected from BM01, BM02 and BM03 to BM1, BM2 and BM3 in lines 424-428, 591, 600, 607  and in Table S1.

Q: Small note: It would be good to discuss the results in the discussion section instead of the results section.

A: In order to avoid too much confusion, we prefer to discuss the detailed data in the Result section itself so the reader can refer directly to the relevant tables. In the Discussion part we revisit the results and discuss them from a more global point of view. We hope this approach is acceptable to the Reviewer and Editor.

Q: The conclusions section looks more like an introduction or an abstract. Please, rewrite and include the outcomes of the study.

A: Indeed, we agree with the reviewer that he conclusions section looks more like an introduction or an abstract. We believe this is due to our choice discussed in the previous question. In the first paragraph of the optional Conclusions section we just set out to conclude with the overall take home message for the relevance of this work in the current global economical and industrial backdrop.  In the second paragraph we summarize the results with sentences like: “The obtained results provide essential basic data that inform the initial choice of tobacco plant species/varieties and extraction conditions when designing efficient processes…” without revisiting the actual outcomes for each distinct compound of interest in order to avoid redundancy with the Results and Discussion. Again, we hope this approach is acceptable to the Reviewer and Editor.

Reviewer 3 Report

The manuscript focuses on less explored aspects of the extraction of secondary metabolites from tobacco, providing results and suggestions for future investigation with sufficient novelty, originality and practical importance. The study has been presented in a comprehensive manner, with adequate justification of the adopted experimental approach and logical progression of analysis. Still, in my opinion, there are some questions and points that need revision before the manuscript is suitable for publication.

Abstract:

The abstract should be revised in order to introduce better the main results from the relevant analyses.

Introduction:

There should be some justification why the authors have chosen to present some of the compounds as members of the respective chemical classes (sections 1.1, 1.2, 1.7), while others have been introduced as single compounds (1.3, 1.4, 1.5, 1.6). Besides, what were the grounds for differentiating between chlorogenic acid and ferulic acid in two separate sections, as they both are phenolic acids and are closely related? It also remains unclear why the authors have paid special attention to linoleic acid in the Introduction (1.6); it is not a very characteristic metabolite in cured tobacco leaves (a much better source of glyceride oil and essential AAs are in fact the tobacco seeds), neither is the only important/bioactive representative of the respective group of compounds (as indicated in fig. 1).

The sentence in lines 272-274 sounds a bit ambiguous, as essential oils are complex mixtures of compounds, not single plant molecules; I would recommend to re-phrase it.

Citation of references is needed to support the statements concerning the advantages of the employed microfractionation technology (lines 290-314). In my opinion, the backgrounds for the experimental approach are well described and discussed in the paper (i.e., the choice of microfractionation technology, its advantages, the following analysis, etc.), but the aim of the study is not so clearly defined at this point.

Results:

Tables 2 and 3 present only 22 out of 24 compounds (#23 and #24 are missing; they are not mentioned in the text, either); the missing data should be added.

In the same tables, as well as in the discussion of results and in the Materials and Methods section, the authors must define which tocopherol exactly was identified.

The authors should provide some explanation why exactly N. rustica variety Bakoum Miena was chosen for the method optimization procedure (section 2.3), because this procedure is based on the data from rapid solubility tests and microfractionation conducted with N. tabacum varieties. Did the authors apply the procedure on the rest of the samples and if so, did they detect the same trends in compound recovery concentrations? Such justification of authors’ choice is needed, because those results have been the grounds to select further M13/BM3 combination for the single solvent extraction and for the profiling of all six Nicotiana varieties in section 2.4.

In my opinion, additional discussion of results is needed in section 2.4, at least because they are also related to the initial content of the respective compounds in the cured tobacco leaves (table 2).

The theoretical yields in Table 6 are very interesting from a practical point of view, but the authors should provide information how was the total dry mass yield per hectare estimated. Do the values represent the recorded dry mass yield of the studied tobaccos (i.e., farmers’ data or experimental data) or have they been cited as typical for the variety based on reference data? (if so, references should be provided).

Materials and methods:

Information about the temperature of extraction should be provided (sections 4.5 and 4.6).

Suggested technical corrections:

Table 3 is referenced before Table 2 (see lines 356 and 380); please, either revise table numbers or change paragraph position.

It would be better to use a uniform designation of compounds that have not been detected in the samples (N/F in tables 2 and 3, and NF in table 4), as well as to describe it below the table.

In table S1, it should be MeOH/H2O (7:3, v/v), not 7:1.

The abbreviations of the solvents should be introduced where first mentioned (see sections 4.3 and 4.4, EtOH/MeOH).

Ref. #100 repeats ref. #4.

Author Response

We would like to sincerely thank the Reviewer for helping us improve the manuscript to better represent the scientific data we collected in our study and for identifying our mistakes, making it more accessible for the intended audience.

Abstract:

Q: The abstract should be revised in order to introduce better the main results from the relevant analyses.

A: Because of the overreaching nature of our results that cover a high number of compounds, we are forced to chose a couple of examples to include in the abstract as updated in line 27.

Q: There should be some justification why the authors have chosen to present some of the compounds as members of the respective chemical classes (sections 1.1, 1.2, 1.7), while others have been introduced as single compounds (1.3, 1.4, 1.5, 1.6).

A: Because of the high number of alkaloids in the study we chose to group them together in the introduction. The others are not necessarily needed to be discussed together. On the other hand under Flavonoids we only have Rutin so we changed the title of the section from “Flavonoids” to “Rutin” in line 112; same for “Neopytadiene” in line 272.

Q: Besides, what were the grounds for differentiating between chlorogenic acid and ferulic acid in two separate sections, as they both are phenolic acids and are closely related?

A: Although we do not wish to group CGA and FA together we understand the point of the reviewer so we moved the two sections together in the introduction, now they are not split by Zeatin. (Important: Track changes were turned off in order for the citations to update safely.)

Q: It also remains unclear why the authors have paid special attention to linoleic acid in the Introduction (1.6); it is not a very characteristic metabolite in cured tobacco leaves (a much better source of glyceride oil and essential AAs are in fact the tobacco seeds), neither is the only important/bioactive representative of the respective group of compounds (as indicated in fig. 1).

A: All compounds were chosen based on two factors. One was the abundance in tobacco plants, the other was the economical importance. Therefore, some compounds, even though present in small quantities in tobacco, were chosen to be investigated as long as they present potential industrial value. The quantity of compounds can be enhanced by using different elicitation methods as chemical, biological or physical.

Q: The sentence in lines 272-274 sounds a bit ambiguous, as essential oils are complex mixtures of compounds, not single plant molecules; I would recommend to re-phrase it.

A: The words “essential oils” were removed from line 276

A: The words “essential oils” were removed from line 276

Q: Citation of references is needed to support the statements concerning the advantages of the employed microfractionation technology (lines 290-314). In my opinion, the backgrounds for the experimental approach are well described and discussed in the paper (i.e., the choice of microfractionation technology, its advantages, the following analysis, etc.), but the aim of the study is not so clearly defined at this point.

A: Two references were given: 97A – a paper, 97B – the link of website of Bicoll introduction for Bicoll successful stories. More might be found for irreversible absorption of normal separation technologies. The microfractionation technology is employed to help for easy and fast pretreatment of crude tobacco samples before content determination, which is in particular useful in the very complex matrices aiming in simplifying analytical methods and increasing the respond in the detectors by enrichment approaches. Therefore the microfractionation technology is highly associated to the article research.  

Q: Tables 2 and 3 present only 22 out of 24 compounds (#23 and #24 are missing; they are not mentioned in the text, either); the missing data should be added.

A: Although in the beginning of the study we did have more compounds of interest, these were removed from all results due to inconsistency. We correct our omission by changing the number from 24 to 22 in Tables 2 and 3.

Q: In the same tables, as well as in the discussion of results and in the Materials and Methods section, the authors must define which tocopherol exactly was identified.

A: Indeed it was alpha-tocopherol that was measured. Corrections were made in lines: 389, 392,  Table2, Table3, Table S2

Q: The authors should provide some explanation why exactly N. rustica variety Bakoum Miena was chosen for the method optimization procedure (section 2.3), because this procedure is based on the data from rapid solubility tests and microfractionation conducted with N. tabacum varieties. Did the authors apply the procedure on the rest of the samples and if so, did they detect the same trends in compound recovery concentrations? Such justification of authors’ choice is needed, because those results have been the grounds to select further M13/BM3 combination for the single solvent extraction and for the profiling of all six Nicotiana varieties in section 2.4.

A: Method optimization was run only on rustica varieties mainly B.M.  We included in the text in lines 428-430: “Method optimization was only conducted on the N. rustica variety Bakoum Miena in order to objectively compare only the effects of the method and solvent tested and exclude differences in matrix effect which were be assessed in section 2.3.”

Q:In my opinion, additional discussion of results is needed in section 2.4, at least because they are also related to the initial content of the respective compounds in the cured tobacco leaves (table 2).

A:This issue is a typical issue of comparing apples and oranges and it’s very important that the reviewer brought it up to better explain in the manuscript. As requested by the reviewer, additional discussion was added in lines 486: “Although the absolute compound recovery values for the single solvent profiling experiment are not directly comparable to the ones referring to the microfractionation coverage in the N. tabacum varieties (Table 2), the trends are similar for most of the compounds. For those compounds where the trends are not similar, the discrepancy could be explained by the inherent difference in the extraction methods (volumes of solvent used, starting material weight, temperature etc.) and technologies applied.”

Q:The theoretical yields in Table 6 are very interesting from a practical point of view, but the authors should provide information how was the total dry mass yield per hectare estimated. Do the values represent the recorded dry mass yield of the studied tobaccos (i.e., farmers’ data or experimental data) or have they been cited as typical for the variety based on reference data? (if so, references should be provided).

A:The data on yields come from PMI internal database based on experimental field cultivation. Description was added to table legend, line 502.

Materials and methods:

Q:Information about the temperature of extraction should be provided (sections 4.5 and 4.6).

A:Extraction was performed at room temperature. The precision was added in lines: 594, 603, 618.

Suggested technical corrections:

Q: Table 3 is referenced before Table 2 (see lines 356 and 380); please, either revise table numbers or change paragraph position.

A:In order to correct this ordering of tables we decided to reference both Tables 2 and 3 in line 359.

Q: It would be better to use a uniform designation of compounds that have not been detected in the samples (N/F in tables 2 and 3, and NF in table 4), as well as to describe it below the table.

A:Formatting was changed to N/F in Table 4.

Q: In table S1, it should be MeOH/H2O (7:3, v/v), not 7:1.

A:The error was corrected in Table S1.

Q: The abbreviations of the solvents should be introduced where first mentioned (see sections 4.3 and 4.4, EtOH/MeOH).

A: Solvents were introduced for the first time in section 2.1. In section 4.4 we removed the redundant solvent names.

Q: Ref. #100 repeats ref. #4.

A: Ref #100 replaced by ref # 4 in line 496. (Important: Track changes were turned off in order for the citations to update safely.)